# Evaluation of cloud height, optical thickness, and phase retrievals from the CHROMA algorithm applied to Sentinel-3 OLCI data

Andrew M. Sayer<sup>1,2</sup>, Brian Cairns<sup>3</sup>, Kirk D. Knobelspiesse<sup>2</sup>, Luca Lelli<sup>4</sup>, Chamara Rajapakshe<sup>2,5</sup>, Scott E. Giangrande<sup>6</sup>, Gareth E. Thomas<sup>7</sup>, and Damao Zhang<sup>8</sup>

Correspondence: Andrew M. Sayer (andrew.sayer@nasa.gov)

Abstract. We previously developed the Cloud Height Retrieval from O<sub>2</sub> Molecular Absorption (CHROMA) algorithm for the Ocean Color Instrument (OCI) on the new NASA Plankton, Aerosol, Cloud, ocean Ecosystem (PACE) mission. Here, we apply CHROMA to observations from the Ocean Land Colour Instrument (OLCI) to guide expectations for PACE, as it will take some time to obtain large-scale validation data for OCI. We use cloud top height (CTH), phase, and (for liquid clouds) cloud optical thickness (COT) data from the ground-based Atmospheric Radiation Measurement (ARM) network to evaluate the OLCI retrievals. We found that OLCI and Moderate Resolution Imaging Spectroradiometer (MODIS) CTH compare similarly well to the ARM reference. OLCI has a tendency to underestimate CTH as CTH increases, and algorithm assumptions about cloud geometric thickness may contribute to this. ARM COT from multifilter shadowband radiometers (MFRSR) and Sun photometers are well-correlated with one another, albeit with a roughly 30% offset on average; OLCI and MODIS COT agree more closely with the MFRSR data. OLCI retrieval uncertainty estimates show skill at telling low-uncertainty cases from highuncertainty ones, although CTH uncertainties are underestimated. Additionally, we compare the OLCI data to satellite retrievals based on thermal infrared measurements from MODIS and and Sea and Land Surface Temperature Radiometer (SLSTR) data. Differences are broadly consistent with physical expectations based on the A-band vs. thermal techniques, although one key challenge in such aggregated comparisons is different cloud masking sensitivities and algorithm failure rates meaning additional sampling differences are introduced. We conclude by discussing the transition to and possible enhancements for PACE OCI.

<sup>&</sup>lt;sup>1</sup>Goddard Earth Sciences Technology And Research (GESTAR) II, University of Maryland Baltimore County, Baltimore, MD, USA

<sup>&</sup>lt;sup>2</sup>NASA Goddard Space Flight Center, Greenbelt, MD, USA

<sup>&</sup>lt;sup>3</sup>NASA Goddard Institute for Space Studies, New York, NY, USA

<sup>&</sup>lt;sup>4</sup>Remote Sensing Technology Institute, German Aerospace Centre (DLR), Wessling, Germany

<sup>&</sup>lt;sup>5</sup>SSAI, Lanham, MD, USA

<sup>&</sup>lt;sup>6</sup>Environmental and Climate Sciences Department, Brookhaven National Laboratory, Upton, NY, USA

<sup>&</sup>lt;sup>7</sup>STFC Rutherford Appleton Laboratory, Didcot, UK

<sup>&</sup>lt;sup>8</sup>Pacific Northwest National Laboratory, Richland, WA, USA

## 1 Introduction

The NASA Plankton, Aerosol, Cloud, ocean Ecosystem (PACE) mission (https://pace.gsfc.nasa.gov) launched on February 8 2024. PACE's primary payload is the Ocean Color Instrument (OCI), a broad-swath (2,600 km) passive imaging radiometer with continuous spectral coverage from the ultraviolet to near-infrared (NIR), seven discrete channels in the NIR and short-wave infrared (SWIR), and an approximately 1.2 km horizontal pixel size at the sub-satellite point (Werdell et al., 2019; Meister et al., 2024). OCI will be used for (among other things) routine generation of a suite of cloud data products: a cloud mask, cloud optical thickness (COT) at mid-visible wavelengths, cloud effective radius (CER), cloud top pressure (CTP), and cloud phase (liquid droplets or ice crystals). Together COT, CER, and phase will also be used to calculate cloud water path (CWP). CTP will also be transformed to cloud top height and temperature (CTH, CTT respectively) with the use of ancillary meteorological profiles. PACE also carries two cubesat-sized multi-angle polarimeters, namely the Hyper-Angle Rainbow Polarimeter 2 (HARP2; Martins et al., 2018) and Spectro-Polarimeter for planetary EXploration (SPEXone; van Amerongen et al., 2019). These are also expected to provide cloud retrieval products, although they will not be further discussed here.

Several OCI cloud products have algorithmic heritage with data from NASA's MODerate resolution Imaging Spectrora-diometer (MODIS) and Visible Infrared Imaging Radiometer Suite (VIIRS) sensors (Platnick et al., 2003, 2021): specifically, the "cloud optical properties" code that generates COT, CER, and derived CWP (Platnick et al., 2017) is being run on OCI. However, unlike MODIS and VIIRS, OCI lacks thermal infrared (TIR) channels which were used for cloud mask, CTH, and phase retrievals for these sensors (e.g. Menzel et al., 2008). As a result, for OCI new cloud mask and phase algorithms are being researched (Coddington et al., 2017). Sayer et al. (2023) developed the Cloud Height Retrieval from O<sub>2</sub> Molecular Absorption (CHROMA) algorithm to fill the other gap: cloud top altitude. That study included simulated retrievals for OCI and the Ocean and Land Colour Instrument (OLCI) on the Sentinel-3 satellites (Donlon et al., 2012), which was identified as a suitable proxy for OCI due to similar spatial resolution and information content in the O<sub>2</sub> A-band that the algorithm uses.

In advance of launch, we applied CHROMA to one month (July 2019) of Sentinel-3A OLCI data globally and for an extended period (October 2016 to December 2022) over select several ground validation sites. These analyses are the subject of the present paper. Sayer et al. (2023) found that OLCI and OCI should provide CTP with similar fidelity, so analysis of retrievals from CHROMA applied to OLCI should provide information useful to guide user expectations of OCI CTP data and assist in algorithm refinements. Section 2 introduces the OLCI instrument and a brief summary of CHROMA (a full description is in Sayer et al., 2023), as well as the other satellite and ground data sets used in this study. Section 3 describes the comparison against ground-based data, and Section 4 the comparison with other satellite retrievals. Finally, Section 5 provides a summary and some possibilities for algorithm refinement on PACE OCI.

## 2 Data sets used in this study



## 2.1 CHROMA applied to OLCI measurements

OLCI is a pushbroom imaging radiometer with 21 spectral channels across the visible and NIR and a swath width of 1,270 km (Donlon et al., 2012). Its "reduced resolution" data mode provides a horizontal pixel size of about 1.2 km, similar to OCI. It is tilted 12.6° westwards to decrease the amount of Sun glint observed over oceans. OLCI flies on a Sun-synchronous polar orbit with a 10 am local solar Equatorial crossing time on the Sentinel-3A (launched 2016) and 3B (launched 2018) satellites; only Sentinel-3A data are used here.

The CHROMA algorithm (Sayer et al., 2023) uses four of OLCI's channels near the O<sub>2</sub> A-band - named OA12 to OA15, with nominal central wavelengths of 753.75, 761.25, 764.38, and 767.50 nm and full width at half maximum (FWHM) of 7.5, 2.5, 3.75, and 2.5 nm respectively. It is a Bayesian retrieval approach using the Optimal Estimation (OE) technique (Rodgers, 2000) which accounts for uncertainties on the measurements and forward radiative transfer assumptions to simultaneously retrieve COT, CTP, and surface albedo (the latter with a significant prior constraint). The principle behind the algorithm is that the observed top-of-atmosphere (TOA) reflectance in the so-called 'window' channel (outside the O<sub>2</sub> A-band) at 753.75 nm is sensitive to COT and surface albedo but not cloud altitude; given a constraint on albedo, the brightness in this channel is then informative on COT. The other three channels see O<sub>2</sub> absorption of varying strengths, and for a given COT and surface albedo their brightness is most strongly dependent on cloud altitude. Other parameters (including within-cloud structure, particle microphysics, and aerosol loading) are assumed and the uncertainty on the TOA reflectance due to these assumptions is included in the retrieval. Advantages of OE include that, if these uncertainties are well-specified, it provides quantitative uncertainty estimates on the retrieved quantities, and a measure of goodness-of-fit (the 'cost' statistic; Rodgers, 2000).

As is common in satellite data processing, we use precalculated lookup tables (LUTs) of radiative transfer results which are iteratively interpolated linearly to find the OE solution. There are two LUTs, one corresponding to simulations for liquid water clouds, and another for ice phase clouds. In both cases clouds are single-layer and single-phase. The algorithm runs twice on each cloudy pixel (once per LUT) and the retrieval with the lower cost statistic (i.e. greater consistency between the measurements and forward model at the solution) is used in subsequent analysis here. PACE OCI will-is expected to have a separate cloud phase product, but none is available for OLCI, which is why we use the lowest-cost method. Based on expected ranges of parameters, COT is retrieved in  $\log_{10}$  space on the interval  $[-0.5, 1.\dot{6}]$ ; CTH (when framed in terms of a standard atmospheric profile) on the intervals [0.3, 7] km for liquid-phase clouds and [3, 17] km for ice-phase clouds; and surface albedo on the interval [0,1] (Sayer et al., 2023). We provide some additional details necessary to go from simulated retrievals as in Sayer et al. (2023) to real measurements:

- We use the cloud mask from the operational OLCI pixel identification algorithm (IdePix), described by Wevers et al. (2021), and only process cloudy pixels with solar zenith angles smaller than 80°.
- We use Modern-Era Retrospective Analysis for Research and Applications, Version 2 (MERRA-2) meteorological profiles to account for variations in atmospheric pressure and convert retrievals between CTP and CTH, and snow/ice cover

status for each pixel (Gelaro and Coauthors, 2017). These data are available on a  $0.5^{\circ}$  latitude by  $0.625^{\circ}$  longitude grid at an hourly cadence.

- The prior surface albedo is obtained from the version 2.1 TROPOspheric Monitoring Instrument (TROPOMI) Lambertian equivalent reflectivity (LER) data base (Tilstra et al., 2024). This provides a climatology of spectral monthly LER on a 0.125° equal-angle grid; we use the 758 nm value (taken as spectrally flat across the four OLCI channels). For each month, two values are tabulated, corresponding to climatologies from snow- or ice- covered vs.—snow- and ice-free scenes (and the MERRA2 data inform which is used for a given pixel). The data base also includes an optional directional modification of LER; as recommended by Tilstra et al. (2024), the directional component is not used here due to the opposite orbital directions of TROPOMI and Sentinel-3. As the data base is spatially coarse, some coasts and islands report a low water-like LER; in these cases, if the data base's LER is under 0.05, the pixel's LER is set to 0.2 with an uncertainty of 0.05 (based on typical values and variability for land scenes in the data base).
- Uncertainty on the prior surface albedo is also provided within the same data base; we set a floor of 0.02 on this to account for sub-grid variability.
  - The contiguous OLCI swath is formed from five cameras; the sensor design leads to an across-track variation of spectral response function commonly known as 'smile' distortion (Preusker, 2021). As a result the retrieval LUTs include two additional dimensions accounting for variation of band centre and FWHM across each camera and with time. Preusker (2021) provide a tool to calculate the effective central wavelength and bandwidth for each OLCI pixel, used here. This is important because the O<sub>2</sub> absorption features are strong and so poor spectral response characterisation can lead to significant biases in retrieved feature (whether cloud, aerosol, or surface) pressure (Lindstrot et al., 2009).
  - For simulated retrievals Sayer et al. (2023) assumed a spectrally-correlated sensor calibration uncertainty (a perfectly correlated error covariance matrix). On-orbit calibration efforts for Sentinel-3A OLCI suggests a radiometric high bias of 2-3 % in the relevant bands (ESA/EUMETSAT, 2023). To account for that, here OLCI TOA reflectances were decreased by 2.5 % for all four bands used, and the remaining calibration uncertainty was taken as 2 % and spectrally uncorrelated (i.e. as a noise term).
  - The first guess at the solution is the LUT node point with the smallest cost function. Pixels which fail to move from this first guess (typically clear-sky pixels erroneously flagged as cloud), or with a retrieval cost of 30 or more (significant misfit with observations, such as incorrect surface model or pixel misclassification; Sayer et al., 2023) are not considered in subsequent analysis.

# 2.2 Ground-based observations from ARM







The US Department of Energy's Atmospheric Radiation Measurement (ARM) program maintains a suite of sensors operating routinely at several permanent facilities, as well as shorter-term (months to a few years) deployments in support of field campaigns (Mather and Voyles, 2016; Ackerman et al., 2016). Observations from various instruments are combined and processed

**Figure 1.** Locations of ground sites used for CTH validation. The site names (including acronym representation) are Ascension Island (ASI); Sierras de Córdoba (COR); Graciosa / East North Atlantic (ENA); Gunnison, Colorado (GUC); Houston, Texas (HOU); North Slope of Alaska (NSA); Southern Great Plains (SGP).

to generate "Value-Added Products" (VAPs, Giangrande et al., 2022), some of which are relevant to validation of satellite cloud products (Shupe et al., 2016). We use seven ARM sites here, shown in Fig. 1 (those operating for around a year or more during period 2016-2022), specifically: Ascension Island (ASI); Sierras de Córdoba (COR); Graciosa / East North Atlantic (ENA); Gunnison, Colorado (GUC); Houston, Texas (HOU); North Slope of Alaska (NSA); Southern Great Plains (SGP). We use datasets from the central or main facilities, contingent on whether this is an ARM fixed (ENA, NSA, SGP) or ARM mobile facility (AMF - the other sites) deployment; information on AMF capabilities is provided by e.g. Miller et al. (2016) and Giangrande et al. (2017). We use the following VAPs:



1. Active Remote Sensing of Cloud Layers using Ka Band ARM Zenith Radars (KAZRARSCL; Clothiaux et al., 2001; Kollias et al., 2016). This multisensor provides cloud layer boundaries (top and base height for up to 10 layers identified in each vertical column) on a vertical grid of 30 m with a 4 s time cadence. Typically, CTH is the 'echo top' height from the ARM Ka band (35 GHZ) radar; however, for shallow and low-COT clouds it is sometimes determined by micropulse lidar (see discussion in e.g. Mechem et al., 2015; Mechem and Giangrande, 2018). We take the CTH uncertainty as the greater of the range of CTH within the averaging window (see below) or the 30 m vertical grid size. This data set is available at all ARM sites used in this study.

- Sun photometer COT (SPHOTCOD), which is based on the 3-channel algorithm of Chiu et al. (2012). This uses solar zenith radiance measurements at visible and SWIR wavelengths and is retrieved for overhead liquid-phase clouds from patchy to overcast conditions on a 5 min time cadence (although this is reported as a mean and standard deviation of higher-frequency measurements). We take the COT uncertainty as the quadrature sum of this standard deviation and the 17 % retrieval uncertainty estimated by Chiu et al. (2010). This data set is available for only the ENA and SGP sites, and was recently extensively analyzed by Sookdar et al. (2025).
  - 3. Multifilter rotating shadowband radiometer (MFRSR) COT (MFRSRCLDOD; Turner et al., 2021), which is based on diffuse transmittance measurements at 415 nm. It is only available for liquid clouds and locally overcast conditions (at least 90 % cloud cover from an effective 160° field of view, inferred from downwelling shortwave irradiances; Long et al., 2006). It is most valid for a COT of approximately 7 or higher (though lower COTs are included in the data set as well), is provided on a 1 min time cadence, and is available for the ASI, COR, ENA, and SGP sites. The data set also includes an estimate of COT uncertainty.





4. Thermodynamic cloud phase (THERMOCLOUDPHASE; Shupe, 2007) takes in multiple VAPs and identifies each cloud layer in KAZRASRCL KAZRARSCL as liquid water droplets, ice crystals, or mixed-phase. This is provided on a 30 s time cadence, and is only available at the high latitude sites (COR and NSA). While CHROMA (and MODIS) do not include mixed-phase cloud retrievals, we retain this category in order to better understand how the satellite retrievals behave in these situations. Additionally, we classify as mixed-phase any column which contains either mixed-phase layers, or contains both liquid and ice-phase layers.

Based on physical understanding and simulated retrievals (Sayer et al., 2023) we expect that retrieval performance will differ between optically-thick single-layer cloud systems (generally seen as the ideal), optically-thin single-layer systems (where there is more sensitivity to assumptions about the underlying surface reflectance), and multi-layer systems (where uncertainty is dependent on the properties of each cloud layer). We therefore consider these three categories separately for aggregate CTH statistics. The PACE mission goal (Werdell et al., 2019) is that 65 % of scenes with a COT≥3 should have a CTP uncertainty 60 or smaller, so we adopt COT=3 as the division for optically-thin cases (using the satellite-retrieved COT), noting that this may include some cases of undetectable sub-pixel broken cloudiness combined with higher-COT clouds. Additionally, we track the fraction of matchups meeting this goal (translating 60 to a CTH using a standard atmospheric profile), as well as the fraction meeting COT uncertainty goals of ±25 % and ±35 % for liquid and ice-phase clouds, respectively ARM is not the only available ground network suitable for CTH validation. Another, Cloudnet, has several dozen sites (Illingworth et al., 2007); we focus on ARM partially because its sites are less geographically clustered (Cloudnet sites are predominantly in Europe) and because it also offers the opportunity to evaluate liquid COT which is not available from Cloudnet. Satellite CTH retrievals have been evaluated using ARM (e.g. Smith et al., 2008), Cloudnet (e.g. Wang and Stammes, 2014; Compernolle et al., 2021), or both (e.g. Sayer et al., 2011; Lelli et al., 2012; Vinjamuri et al., 2023) in similar ways. As the OLCI implementation of CHROMA is a research-level code it was not feasible to process data from both networks, and we felt that ARM alone provided sufficient

data for an intial evaluation of CHROMA. For eventual application to routine PACE OCI data, we intend to also evaluate at Cloudnet sites.

# 160 2.3 MODIS cloud retrievals

We use MODIS Terra data due to the similarity between Terra and Sentinel-3 overpass times (10:30 and 10 am local solar Equatorial crossing for daytime nodes, respectively) and known diurnal patterns of cloud property variations which would confound comparisons if Aqua (1:30 pm local time) were used instead.

The MODIS cloud products used here come from two separate algorithms, with a pixel size of 1 km at the sub-satellite point. Cloud altitude is obtained from TIR bands in atmospheric windows near 11 and 12 µm and so-called CO<sub>2</sub> slicing bands centred from 13 to 14 µm, which provide additional sensitivity to optically-thin high clouds (Baum et al., 2012). This data set does not provide pixel-level uncertainty estimates. MODIS COT is obtained from a bispectral approach (Platnick et al., 2017), combining a band at which clouds are (almost) purely scattering (typically 650 or 865 nm) with a SWIR band where cloud droplets absorb strongly to simultaneously infer COT and CER as the two measurements are roughly orthogonal to these properties. This data set provides pixel-level uncertainty estimates. Cloud phase is determined through a series of tests based mostly on retrieved CER and CTT. Finally, MODIS has a flag to indicate the suspected presence of multi-layer clouds, which is determined from a combination of solar vs. thermal cloud phase estimates, precipitable water retrievals, and spectral reflectance/temperature tests (Marchant et al., 2020).

## 2.4 SLSTR cloud retrievals



The Sea and Land Surface Temperature Radiometer (SLSTR) flies alongside OLCI on the Sentinel-3 satellites (Donlon et al., 2012), and is a successor to the Along-Track Scanning Radiometer (ATSR) sensor series. The instrument has a nadir-looking and an aft view and measures from visible to TIR wavelengths with a pixel size of 0.5 or 1 km (dependent on band).

We use data from the Optimal Retrieval of Aerosol and Cloud (ORAC) algorithm which was previously applied to the ATSR sensors (Sayer et al., 2011; Poulsen et al., 2012) and others (Sus et al., 2018). ORAC (like CHROMA) is an OE approach which simultaneously determines COT, CER, and height from a combination of solar and thermal bands (McGarragh et al., 2018). Cloud phase is determined using spectral tests described by Pavolonis and Heidinger (2004) and Pavolonis et al. (2005). The Copernicus Climate Change Service (C3S) version 3.1.1 data set we use (Poulsen et al., 2022) is provided at 0.1° resolution on a daily basis.

# 3 Evaluation against ARM data

# 185 3.1 Matchup methodology

As clouds can vary rapidly in space and time, we use fairly strict criteria for a successful matchup to decrease the contribution of spatiotemporal mismatch to apparent disagreement. As the ARM data are upward-looking while the satellite data are mostly

off-nadir, we perform a parallax correction to account for the apparent shift in cloud horizontal location from the satellite's point of view. This is necessary even for a single-viewing instrument as the satellite data are natively geolocated to surface level, so there will be a distortion unless the satellite is viewing directly at nadir. This is only an approximate correction due to uncertainties in retrieved CTH the satellite CTH retrieval and unknown vertical extent. First, neglecting local curvature (a negligible effect on these scales), the total horizontal shift (in km)  $\Delta$  is estimated as







$$\Delta = h \tan(\theta_{\rm v}) \tag{1}$$

where h is the retrieved CTH (retrieved by satellite) and  $\theta_v$  the viewing zenith angle in degrees, such that  $\theta_v = 0^\circ$  is a nadir view and  $\theta_v = 90^\circ$  is a limb view. This shift  $\Delta$  can be split into longitudinal ( $\Delta_x$ ) and latitudinal ( $\Delta_y$ ) components:

$$\Delta_{\mathbf{x}} = \Delta \sin(\phi_{\mathbf{v}}) \tag{2}$$

$$\Delta_{y} = \Delta \cos(\phi_{y}) \tag{3}$$

Here,  $\phi_v$  is the viewing azimuth angle in degrees, measured clockwise from North, from the satellite pixel location to the subsatellite point. By convention this is reported in the interval  $[-180^{\circ}, 180^{\circ}]$  in which case scenes to the east of the sub-satellite point along the scan line have negative azimuth and those to the west have positive; due to the periodicity of the  $\sin$  and  $\cos$  functions, it mathematically does not matter whether this convention or the range  $0^{\circ}$  to  $360^{\circ}$  is used. Assuming a spherical Earth with 110 km per degree latitude and  $110\cos\gamma$  km per degree longitude (where  $\gamma$  is the latitude), we subtract these shifts from the apparent satellite geolocation (as we wish to undo the parallax-induced distortion) to estimate the true latitude and longitude of the clouds the satellite is seeing.

For both the satellite and ARM data, we use medians calculate medians (rather the means) for the spatiotemporal average to reduce the impact of outliers (that can occur due to e.g. multiple cloud systems within the averaging zone). For cloud phase, we take the modal classification. Satellite pixels with centres within  $\pm 2 \,\mathrm{km}$  of the ARM site, and ARM measurements within  $\pm 2.5 \,\mathrm{min}$  of the satellite overpass time, are considered (except for Sun photometer COT where we use  $\pm 5 \,\mathrm{min}$  due to its lower measurement frequency). Additionally, we exclude points where the range of CTH in the ARM data is 1 km or more, as here sampling-related differences are likely more severe. The focus of this analysis is on OLCI-ARM comparisons. However, as MODIS is a well-used satellite product and to provide a point of comparison to OLCI results (which has not, to our knowledge, been reported in the literature) we also provide MODIS-ARM comparisons.

Based on physical understanding and simulated retrievals (Sayer et al., 2023) we expect that retrieval performance will differ between optically-thick single-layer cloud systems (generally seen as the ideal), optically-thin single-layer systems (where there is more sensitivity to assumptions about the underlying surface reflectance), and multi-layer systems (where uncertainty is dependent on the properties of each cloud layer). We therefore consider these three categories separately for aggregate CTH statistics. The PACE mission goal is that 65% of scenes with a COT>3 should have a CTP uncertainty 60 mb or smaller (Werdell et al., 2019), so we adopt COT=3 as the division for optically-thin cases (using the satellite-retrieved COT), noting that this may include some cases of undetectable sub-pixel broken cloudiness combined with higher-COT clouds. Additionally, we track the fraction of matchups meeting this goal, translating 60 mb to a CTH using a standard atmospheric

Figure 2. Cumulative frequency distributions of ARM CTHs, for the ARM-OLCI matchups. Different colours indicate different sites.

profile (Dubin et al., 1976). This leads to about a 1 km uncertainty goal for a CTH of 3 km, and about 3 km for a CTH of 12 km. We also record the fraction meeting COT uncertainty goals of  $\pm 25$  % and  $\pm 35$  % for liquid and ice-phase clouds, respectively.

## 3.2 CTH comparisons





A statistical summary of the OLCI CTH validation results are shown for each site and category in Table 1; the equivalent for MODIS are in Table 2. We describe and focus on the importance of these metrics in the discussions below. In general, we focus on robust metrics such as Spearman's rank correlation coefficient over Pearson's linear correlation coefficient, and median rather than mean bias. Some variability in retrieval performance between sites is expected as ASI and ENA are islands while NSA is snow-covered and at large solar zenith angles; in Sayer et al. (2023) we found, based on simulated retrievals, that (compared to land) errors should be smaller for ocean scenes and larger for snow- or ice-covered scenes. This fact is particularly relevant when examining bulk comparison statistics since (due to its high latitude and so more frequent overpasses), the NSA site is disproportionately over-represented in the set of available provides over half of all matchups. Additionally, ASI, COR, GUC, and HOU were temporary ARM sites not available for the whole analysis period (hence fewer matchups).

While sampling limits the conclusions—and the available sites are not globally-representative—we can state some general tendencies. As expected, errors (whether sign or magnitude errors) are smaller for single-layer opaque clouds than either the thin cloud or multi-layer subsets. They also tend to be smaller for the ocean sites and larger for NSA. In general, for single-layer opaque clouds, OLCI has slightly smaller median absolute error (MAE) but larger root mean square error (RMSE) than MODIS, implying—which implies the present of more or larger outliers in the OLCI data set. At the island sites (ASI and ENA) and NSA, though, OLCI has a lower RMSE. Some of this can be explained by the underlying sensitivities of the two techniques: A-band retrievals are sensitive to pressure contrast (i.e. rate of change of pressure with respect to height) which is greatest in the boundary layer (Sayer et al., 2023); thermal retrievals are sensitive to temperature contrast which is greatest in

**Table 1.** OLCI CTH validation summary statistics for each site for the three analysis categories.  $F_{60}$  and  $F_{\rm ED}$  are the fraction of matchups meeting the 60 mb goal and agreeing within the expected discrepancy, respectively (the latter discussed in Section 3.3).  $R_{\rm S}$  is Spearman (rank) correlation coefficients. Matchup count, median absolute error (MAE), and root mean square error (RMSE) are also shown. Metrics are not reported for categories with fewer than 10 matchups.

| Site    | Count     | $F_{60}$  | $F_{ m ED}$ | $R_{\mathrm{S}}$ | Median bias (km) | MAE (km) | RMSE (km) |
|---------|-----------|-----------|-------------|------------------|------------------|----------|-----------|
| COT 

**Figure 3.** OLCI CTH error as a function of ARM CTH. Diamonds and lines show bin median and central 68 % of data, respectively. Data are split into COT<3 (grey), single-layer COT≥3 (blue), and multi-layer COT≥3 (red) cases. Numbers in parentheses indicate counts for each category. The dashed black line indicates zero error.

Additionally, in both the OLCI and MODIS data sets, the RMSE is inflated due to a few extreme outliers. Manual examination of these scenes (not shown) suggests that some are not true retrieval errors, but rather spatio-temporal mismatch when there are both high and low clouds in the vicinity of a site (but only one type lies above the ARM site, while the other is more common in the area of satellite observations considered). The fairly strict matchup criteria cannot totally eliminate this type of sampling mismatch, and this is a motivation for the aforementioned use of robust metrics such as MAE and rank correlation.





The fraction of matchups meeting the goal uncertainty of 60 mb for single-layer opaque clouds ( $F_{60}$ ) is 0.53, but varies from 0.33 to 0.74 dependent on site. This falls short of the goal fraction of 0.65 on average. While sampling is limited at some sites, of the sites with at least 100 matchups, the ocean sites appear to perform a bit better than the others.  $F_{60}$  is lower for optically-thin or multi-layer cloud systems. This is all in line with our simulated retrievals in Sayer et al. (2023). For MODIS, the equivalent values are 0.35-0.76 for single-layer opaque clouds<del>overall</del>, indicating similar performance. The site-aggregated overall value is 0.49, lower than OLCI, but the proportion of matchups at NSA is higher in MODIS than OLCI so this number is less directly comparable. MODIS also performs better for optically-thin and multi-layer systems, likely due to its greater sensitivity to the higher-altitude ice-phase clouds more common in these cases.

Figure 3 examines the error tendencies of the OLCI retrievals as a function of CTH. Here and in later plots, the data are binned with the number of bins equal to the cube root of the number of points (to balance the statistical robustness of each bin with the range of CTH covered by the bin values). Bin-median values are given on both axes, and vertical bars indicate the central one standard deviation (16<sup>th</sup> to 84<sup>th</sup> percentile). We choose this method of presentation as the errors may be asymmetric and skewed, which would be hidden by a mean/standard deviation analysis.

For each of the three categories, the error becomes more negative and more variable as CTH increases. The increased error variability is linked to the aforementioned higher pressure sensitivity in the lower atmosphere than upper (i.e. the same CTP




**Figure 4.** OLCI CTH error as a function of ARM cloud fractional geometric depth (FGD), for clouds retrieved as (a) liquid-phase and (b) ice phase. Diamonds and lines show bin median and central 68 % of data, respectively. Data are split into COT<3 (grey), single-layer COT≥3 (blue), and multi-layer COT≥3 (red) cases. Numbers in parentheses indicate counts for each category. The horizontal dashed black line indicates zero error, and the vertical dashed black lines indicate the retrieval's FGD assumption for each cloud phase.

uncertainty translates to a larger CTH uncertainty higher in the atmosphere). As NSA is overrepresented in the sample and contributes the most matchups of any single site and is known to be a more difficult site for retrievals, we also created an equivalent of Fig. 3 excluding it (not shown). This has smaller magnitude errors but the same general tendency of increasing negative bias with increasing CTH, suggesting the tendency is robust between sites and not primarily a feature at NSA. One exception is the overestimate of CTH of low clouds for COT<3 conditions became significantly smaller. This is likely due to these being low-level mixed-phase clouds, which are common in the Arctic (e.g. Fig. 2 and Shupe et al., 2006; Shupe, 2011), retrieved by OLCI as being ice-phase (for which CHROMA permits a minimum CTH of 3 km).

The negative bias in CTH suggests some systematic error in OLCI calibration, and/or in the retrieval radiative transfer (e.g. modeling of O<sub>2</sub> absorption) or assumptions (discussed shortly); while the ARM data may be sensitive to lower cloud droplet concentrations (and retrieve a higher top) than satellite remote sensing, this would not account for a multi-km error. Clouds being retrieved lower down in the atmosphere than they are implies insufficient absorption in the forward model. One potential source of this would be limitation of the HITRAN absorption line data base (Gordon et al., 2022) and/or MT\_CKD continuum models (Mlawer et al., 2012) used in the retrieval. We use the latest HITRAN 2020 line data base and Voigt line shape (Gordon et al., 2022); while more complicated line shape models have been developed and shown to be useful for the O<sub>2</sub> A-band (e.g. Tran and Hartmann, 2008; Mendonca et al., 2019), the parameters for these shapes are not available in the present version of HITRAN. Additionally, it is not clear how large or systematic any differences from Voigt would be integrated over the OLCI (or OCI) spectral response.

In Sayer et al. (2023) we identified that a key algorithmic assumption influencing CTH bias was the assumed cloud geometric thickness. Given a CTH and a cloud base height (CBH), we defined cloud fractional geometric depth FGD = 1 - (CBH/CTH) such that FGD=1 indicates full cloudiness vertically from the cloud top to the surface, and FGD=0 an infinitesimally geometrically thin cloud concentrated at the cloud top. Previous research had found that FGD was not robustly retrievable from OLCI

**Figure 5.** (a) Median and (b) range of central 68.4 % of values of cloud FGD from ARM, binned as a function of OLCI-retrieved COT and ARM CTH. Bins with fewer than five matchups contributing are shaded grey.

measurements in many cases (Fischer and Preusker, 2021), so we opted to use fixed values for this initial version, of FGD=0.5 for liquid clouds and 0.25 for ice clouds, based on examination of ARM data (Kollias et al., 2016), but noted (Section 3.2.3 and Fig. 4 of Saver et al., 2023) that there was considerable variability in FGD as a function of CTH and by site.





Our simulated retrievals showed (Section 4.1.2 and Fig. 13 of the above) that a true FGD larger than assumed resulted in an overestimate of retrieved CTP (i.e. underestimate of CTH), and vice versa. Although the magnitude of this error depended on multiple factors it could exceed the 60 mb goal uncertainty, which corresponds to about 1 km for a CTH of 3 km, and about 3 km for a CTH of 12 km. Figure 4 helps assess whether the retrievals on real data show similar tendencies. One difficulty is that due to the absence of a ground-truth cloud phase at most of the ARM sites we rely on the satellite-retrieved phase, and that is expected to be uncertain for low-COT or multi-layer cases (Sayer et al., 2023). Nevertheless, this figure shows near-zero CTH bias for cases where the FGD is similar to the assumption, and a tendency for increasing negative bias as FGD increases (except for, potentially, the optically-thin ice clouds case).

Figure 5 shows the high variability of FGD from the ARM data, as a function of CTH (from ARM) and COT (from OLCI due to the lack of a ground-truth). While Fig. 5(a) reveals some general tendencies, Fig. 5(b) shows a large degree of variability for a given COT/CTH combination, and the available data do not robustly span the entire range of parameter space. In general increasing COT is associated with larger FGD, which is consistent with expectations for convective growth of clouds. For optically-thin clouds, higher CTH (e.g. cirrus) tend to have smaller FGD than lower CTH (e.g. low cumulus or stratus) clouds. On average, however, values tend to be slightly higher than CHROMA's current assumption of 0.5 for liquid-phase clouds and 0.25 for ice-phase clouds. One possibility for OCI is to update CHROMA's assumption about FGD with data derived (and extrapolated) from Fig. 5. Alternatively, these could be used as *a priori* values and uncertainties if FGD were to be retrieved rather than fixed. However, getting the right FGD alone is not a guarantee of an unbiased retrieval because the within-cloud

extinction profile may also differ from what is assumed. Additionally, the data gathered from these limited sites may not be globally representative, and uncertainties in the COT data may counfound confound the true relationships.

Although other retrievals based on O<sub>2</sub> absorption have also found negative offsets in retrieved CTH compared to thermal or active remote sensing techniques (e.g. Sneep et al., 2008; Lelli et al., 2012; Compernolle et al., 2021), we do not believe this is an inherent problem with O<sub>2</sub>-based retrievals. Rather, all these retrievals make (different) simplifying assumptions about clouds. Some approaches (e.g. Sneep et al., 2008) treat clouds as a Lambertian reflector, meaning they neglect the potential for multiple scattering inside the clouds. This naturally leads to the algorithm retrieving a lower altitude than the actual top. Other algorithms which incorporate within-cloud scattering (Lelli et al., 2012; Saver et al., 2023) tend to show smaller biases because they model the cloud more realistically. However, they are still biased if they make inappropriate assumptions for a specific cloud. For all algorithm types, these assumptions may be violated more significantly for higher clouds (as, by nature, they offer more opportunity for variation in vertical structure). We also note that retrievals with higher information content offer the potential to retrieve additional parameters related to cloud structure (e.g. CBH) and ameliorate some of these biases (see e.g. theoretical work by Heidinger and Stephens, 2000). Richardson et al. (2019) presented a technique to retrieve cloud geometric thickness of marine stratocumulus from Orbiting Carbon Observatory 2 (OCO-2) A-band measurements, which have a finer spectral resolution than OLCI, using Cloud-Aerosol Lidar and Infrared Pathfinder Satellite Observations (CALIPSO) profiles to provide a prior constraint on top pressure. Yang et al. (2021) present another OCO-2 method which does not require such a prior. As another example, Nagao et al. (2025) combine O<sub>2</sub> and TIR measurements to retrieve CTH and CBH. Further, although not in all conditions, Fischer and Preusker (2021) found that this was possible in some cases using OLCI A-band measurements alone. These examples raise promise for PACE OCI because in addition to the A-band it also samples the B-band and strong absorption features associated with H<sub>2</sub>O.

Scene heterogeneity and 3D radiative transfer effects (both sub-pixel and horizontal photon transport between pixels) are another potential source of error. Heidinger and Stephens (2002) studied this using radiative transfer modeling and Landsat data, although they focused on the effects on COT retrievals rather than on CTH. In general, studies of heterogeneity have focused mostly on COT and CER (e.g. Zhang and Platnick, 2011). To our knowledge the effect on retrieved height has not been investigated thoroughly and it is not clear whether heterogeneity would result in a random or systematic error in CTH. Whatever the major contributing causes, this CTH-dependent bias has some implications for the overall statistics in Table 1. For example, while 53 % of single-layer opaque clouds meet the 60 mb goal error metric overall ( $F_{60}$ ), the fraction is 0.65 for the subset of them where ARM CTH is 3 km or below and only 0.29 for the subset where ATM CTH is above 3 km. Similarly, the median CTH bias is -0.16 and -1.93 km for these two subsets, vs. -0.37 km overall. Similar variations in performance between lower and higher clouds are also seen in the optically-thin and multi-layer subsets. Our take-away message from this is that mitigating the CTH-dependent retrieval bias should be the main priority for algorithmic refinement.

# 340 3.3 CTH uncertainty evaluation







Finally, the OE technique provides estimates of the uncertainties on the retrieved quantities (Rodgers, 2000), which can be evaluated. We follow our previous methodology for this (Sayer et al., 2020). In essence, OE provides the maximum *a posteriori* 

Figure 6. Expected OLCI-ARM CTH discrepancy vs.  $68^{\rm th}$  percentile of CTH error (see text). Data are split into COT<3 (grey), single-layer COT $\geq$ 3 (blue), and multi-layer COT $\geq$ 3 (red) cases. Numbers in parentheses indicate counts for each category. The dashed black line indicates 1:1.

retrieval solution and an uncertainty assuming Gaussian (Normal) error distributions. We can therefore define an expected discrepancy (ED) between retrievals and ARM data as the quadrature sum of this retrieval uncertainty and the ARM uncertainty (assuming they are independent). Then, statistically, if the uncertainty estimates are well-calibrated then normalised retrieval errors (i.e. retrieval error divided by ED) should follow a Normal distribution with mean 0 and variance 1. In this case the mean is negative because of the negative bias seen in the retrievals. But a corollary of this is that one standard deviation (i.e. ~68 %) of matchups should agree within this ED. Table 1 shows that this is almost met for the COT<3 subset of data (66 %), but not for the single-layer opaque (57 %) or multi-layer (15 %) cases; the retrieval is therefore overconfident. This is not surprising for multi-layer cases because a core assumption of the algorithm (that the cloud system is single-layer) is violated so the OE estimates cease to be applicable (see also discussion in Povey and Grainger, 2015, on conditions required for uncertainty estimates to be meaningful). Interestingly, at the island sites (ASI and ENA) this fraction is exceeded for single-layer opaque clouds implying the retrieval is slightly underconfident. Potentially, if the assumed uncertainty on the *a priori* TROPOMI surface albedo is too low, it could lead to the retrieval overestimating how well-constrained the COT and CTH are.

A related question is: does the algorithm have skill at telling low-uncertainty cases from high ones? This can also be examined statistically (Sayer et al., 2020), shown here in Fig. 6. If data are stratified by ED then, under Gaussian errors, within each bin the 68<sup>th</sup> percentile of observed error should match the ED and the data should fall along the 1:1 line. This is performed with binned data and not individual points because the observed error is a draw from an uncertainty distribution, i.e. for a given level of uncertainty (ED) one needs a distribution of observed errors to assess whether the observations are consistent with expectation. In Fig. 6, for the single-layer opaque subset the observed errors are well-correlated with ED, indicating the

**Figure 7.** MFRSR vs. Sun photometer COT at the ENA (blue) and SGP (green) ARM sites, coincident with MODIS matchups. Numbers in parentheses indicate counts for each site. Horizontal and vertical error bars indicate the uncertainties provided by each ARM data set. The dashed black line indicates 1:1.

uncertainty estimates are skillful. The points mostly lie above the 1:1 line, consistent with the overall slight overconfidence. For the thin and multi-layer subsets, however, the errors and ED do not show this linear agreement indicating that the uncertainty estimates are not skillful. The lack of skill for optically-thin cases traces in part due to the positive bias of low clouds in Figure 3, which are sometimes retrieved as ice phase near 3 km: such category errors are not well captured by the OE uncertainty propagation methodology.

These metrics also depend on the ARM CTH:  $F_{\rm ED}$  for single-layer opaque clouds increases from 0.57 overall to 0.71 where ARM CTH is 3 km or lower, and drops to 0.26 where ARM CTH is above 3 km. For the optically-thin clouds, the equivalent  $F_{\rm ED}$  are 0.66 overall vs. 0.89 and 0.17 for lower and higher clouds, respectively. These metrics should ideally not show any dependence on the true CTH. The limitations of uncertainty estimate skill imply non-negligible contributions to the retrieval error which are not adequately accounted for by the retrieval's uncertainty budget. These may be similar factors contributing to the CTH-dependent bias of retrieved CTH seen in Figure 3. Decreasing this retrieval bias (whether through improved calibration, radiative transfer, or treatment of FGD) might therefore also improve the skill of the uncertainty estimates.

The MODIS CTH product does not provide pixel-level uncertainty estimates so a equivalent evaluation is not possible.

## 3.4 COT comparisons



As discussed in Section 2.2, ARM has liquid COT data from two techniques: MFRSR and Sun photometry. These are independent techniques with different assumptions, and fortunately both are available at the ENA and SGP sites. Fig. 7 compares these two data sets, for the data coincident with MODIS matchups (results for OLCI are numerically similar but with a smaller data

volume). This shows that the two ARM COT data sets are well-correlated with one another but MFRSR COTs are about 30 % lower, with the interquartile range (IQR) of ratios from 0.57 to 0.85. This 30 % offset between the two is consistent between sites and larger than the PACE goal COT uncertainty for liquid clouds of 25 %, which presents a challenge in terms of using these to evaluate satellite retrievals as it is not clear which is more accurate. Interestingly, a similar analysis for SGP by Chiu et al. (2010) did not find such an offset, although their MFRSR COT was retrieved independently from the ARM operational algorithm (McFarlane and Shi, 2012). Additionally, a comparison of MFRSR retrievals vs. aircraft cloud probes from eight profiles at SGP by Min et al. (2003) found a higher level of agreement in COT. An earlier study by Min and Harrison (1996) did find MFRSR COT higher by a factor of  $\sim$ 2 than geostationary retrievals for thick (COT>10) clouds, but due to the older instruments and retrievals used it is not clear whether those results would be reflective of current data. The offset seen in our results is, however, consistent with the recent analysis by Sookdar et al. (2025) who also compared (for a more restrictive subsetting to stratocumulus clouds without ground-level precipitation) Sun photometer and MFRSR COT at the ENA and SGP sites.






Importantly, however, both instruments (and the satellite data sets) have different limitations which affect their quality and appropriateness. Sun photometers have a smaller field of view (1.2°) so are less affected by broken cloudiness (they either see a cloud, or a gap) while the MFRSR technique requires more fully-overcast scenes (see earlier). Satellite pixels are larger so even more sensitive to sub-pixel heterogeneity (e.g. Jones et al., 2012; Loveridge and Di Girolamo, 2024) (Jones et al., 2012; Loveridge and Di Girolamo, 2024). Cloud inhomogeneity and edges can also affect the retrievals in different ways. We have also not applied additional filtering for precipitation beyond ARM standard quality checks. For these reasons, we report results but do not draw as strong conclusions from the COT analysis of this study.

Summary statistics comparing OLCI and ARM COTs are given in Table 3, and comparing MODIS and ARM COTs in Table 4. These are split into single-layer and multi-layer cases using the KAZRARSCL data set; both MFRSR and Sun photometer retrievals should be valid in multi-layer conditions, provided all layers are liquid phase. Note that, unlike for CTH, the MODIS COT algorithm does provide pixel-level uncertainty estimates.

For the MFRSR matchups, ASI is an outlying site where the retrieved OLCI COT is significantly lower than the rest (median ratio 0.47). This might be due to poor surface albedo constraints; Ascension Island where this site is located does not appear in the TROPOMI LER data base used due to its small size (the minimum LER technique used means it is represented by water-like values), meaning CHROMA uses to a default *a priori* value (Section 2.1). Excluding this, the median ratio for the other sites is 0.89 to 1.20 indicating small overall bias. Ratios are lower for multi-layered cases, and for the Sun photometer matchups. The OLCI COTs appear more consistent with the MFRSR data than with the Sun photometer data, both in terms of median ratio and the fraction agreeing within ED being nearer to 0.68. Similar comments apply to the MODIS matchups in Table 4: ratios are closer to 1 for MFRSR than for Sun photometry, and ASI data are a low outlier (though it is unclear whether the reason is surface albedo). MODIS COT uncertainties appear highly underestimated, with  $F_{\rm ED}$  much lower than 0.68. Both data sets are similarly well-correlated with the ground-based COTs, although the limited sample sizes means correlation comparisons should not be over-interpreted (Schönbrodt and Perugini, 2013).

**Table 3.** OLCI COT validation summary statistics for each site for the three analysis categories.  $F_{25}$  is the fraction of matchups meeting the 25 % difference goal; the median ground:satellite COT ratio and interquartile range (IQR) of the ratio are also shown. Other metrics are as in Table 1.

| Site         | Count          | $F_{25}$ | $F_{ m ED}$ | $R_{\rm S}$ | Median ratio | IQR of ratio |  |  |  |
|--------------|----------------|----------|-------------|-------------|--------------|--------------|--|--|--|
| MFRS         | MFRSR matchups |          |             |             |              |              |  |  |  |
| Single-layer |                |          |             |             |              |              |  |  |  |
| All          | 460            | 0.33     | 0.67        | 0.73        | 0.80         | (0.49, 1.18) |  |  |  |
| ASI          | 123            | 0.15     | 0.43        | 0.66        | 0.47         | (0.28, 0.65) |  |  |  |
| COR          | 14             | 0.43     | 0.79        | 0.93        | 1.20         | (0.81, 1.38) |  |  |  |
| ENA          | 188            | 0.41     | 0.80        | 0.74        | 0.89         | (0.60, 1.23) |  |  |  |
| SGP          | 135            | 0.38     | 0.67        | 0.84        | 1.02         | (0.65, 1.54) |  |  |  |
| Multi-l      | layer          |          |             |             |              |              |  |  |  |
| All          | 230            | 0.33     | 0.54        | 0.70        | 0.67         | (0.45, 1.07) |  |  |  |
| ASI          | 29             | 0.14     | 0.38        | 0.33        | 0.41         | (0.23, 0.63) |  |  |  |
| COR          | 7              | -        | -           | -           | -            | -            |  |  |  |
| ENA          | 114            | 0.35     | 0.58        | 0.66        | 0.70         | (0.45, 1.03) |  |  |  |
| SGP          | 80             | 0.36     | 0.52        | 0.71        | 0.75         | (0.53,1.22)  |  |  |  |
| Sun ph       | otometer       | matchu   | ps          |             |              |              |  |  |  |
| Single-      | layer          |          |             |             |              |              |  |  |  |
| All          | 46             | 0.54     | 0.65        | 0.76        | 0.76         | (0.36, 0.98) |  |  |  |
| ENA          | 24             | 0.46     | 0.54        | 0.55        | 0.71         | (0.31, 0.98) |  |  |  |
| SGP          | 22             | 0.64     | 0.77        | 0.73        | 0.82         | (0.55, 0.98) |  |  |  |
| Multi-l      | Multi-layer    |          |             |             |              |              |  |  |  |
| All          | 45             | 0.38     | 0.49        | 0.67        | 0.51         | (0.32, 0.79) |  |  |  |
| ENA          | 28             | 0.46     | 0.54        | 0.47        | 0.50         | (0.30, 0.99) |  |  |  |
| SGP          | 17             | 0.24     | 0.41        | 0.62        | 0.56         | (0.39, 0.65) |  |  |  |

Figure 8 evaluates uncertainty estimate skill for the OLCI COT retrievals. For single-layer cases, uncertainty estimates appear skillful and well-calibrated. This contrasts with the simulated retrievals in Sayer et al. (2023), where we found that COT uncertainty was overestimated - the difference may be due to uncertainties in the real-world data and retrievals not captured by the simulation study. Multi-layer COT uncertainties also appear skillful, but slightly underestimated. The fact that the uncertainty estimates show good skill for both data sets despite the significant offsets between MFRSR and Sun photometer COTs implies that the bulk of the disagreement/error is systematic (i.e. calibration of one or all sensors/algorithms) rather than noise. A better understanding of (and, if necessary, correcting for) the offsets between the ground data would enable better understanding of the satellite retrievals.

Table 4. As Table 3, except for MODIS COT.

| Site                    | Count          | $F_{25}$ | $F_{ m ED}$ | $R_{\rm S}$ | Median ratio | IQR of ratio |  |  |  |  |
|-------------------------|----------------|----------|-------------|-------------|--------------|--------------|--|--|--|--|
| MFRS                    | MFRSR matchups |          |             |             |              |              |  |  |  |  |
| Single                  | -layer         |          |             |             |              |              |  |  |  |  |
| All                     | 656            | 0.39     | 0.16        | 0.79        | 0.82         | (0.52,1.13)  |  |  |  |  |
| ASI                     | 72             | 0.29     | 0.08        | 0.68        | 0.62         | (0.39, 0.83) |  |  |  |  |
| COR                     | 42             | 0.29     | 0.14        | 0.81        | 0.94         | (0.57, 1.43) |  |  |  |  |
| ENA                     | 309            | 0.43     | 0.13        | 0.71        | 0.86         | (0.57,1.12)  |  |  |  |  |
| SGP                     | 233            | 0.40     | 0.24        | 0.84        | 0.83         | (0.50, 1.18) |  |  |  |  |
| Multi-                  | layer          |          |             |             |              |              |  |  |  |  |
| All                     | 445            | 0.29     | 0.08        | 0.64        | 0.63         | (0.39, 0.95) |  |  |  |  |
| ASI                     | 25             | 0.24     | 0.08        | -0.07       | 0.44         | (0.22, 0.68) |  |  |  |  |
| COR                     | 13             | 0.15     | 0.00        | 0.20        | 0.60         | (0.36,1.15)  |  |  |  |  |
| ENA                     | 237            | 0.26     | 0.06        | 0.58        | 0.60         | (0.37, 0.95) |  |  |  |  |
| SGP                     | 170            | 0.35     | 0.11        | 0.75        | 0.70         | (0.44,1.01)  |  |  |  |  |
| Sun photometer matchups |                |          |             |             |              |              |  |  |  |  |
| Single                  | -layer         |          |             |             |              |              |  |  |  |  |
| All                     | 72             | 0.38     | 0.19        | 0.62        | 0.62         | (0.35, 0.85) |  |  |  |  |
| ENA                     | 40             | 0.38     | 0.25        | 0.49        | 0.62         | (0.35, 0.87) |  |  |  |  |
| SGP                     | 32             | 0.38     | 0.12        | 0.69        | 0.66         | (0.39, 0.81) |  |  |  |  |
| Multi-                  | layer          |          |             |             |              |              |  |  |  |  |
| All                     | 73             | 0.21     | 0.14        | 0.57        | 0.41         | (0.26, 0.73) |  |  |  |  |
| ENA                     | 37             | 0.19     | 0.08        | 0.36        | 0.33         | (0.22, 0.70) |  |  |  |  |
| SGP                     | 36             | 0.22     | 0.19        | 0.65        | 0.51         | (0.32,0.87)  |  |  |  |  |

## 420 3.5 Phase comparisons


At present, ARM cloud phase VAPs are only available at two of the sites used in this study: the NSA permanent site near Barrow, Alaska, USA; and the COR site which was a mobile facility deployment in the Sierras de Córdoba mountains in Argentina as part of the Cloud, Aerosol, and Complex Terrain Interactions (CACTI) field campaign for seven months in 2018-2019 (Varble et al., 2021). This means that the results from these sites may not be generalisable to broader conditions; data volume at COR is also low because of the limited CACTI campaign length. Confusion matrices for OLCI and MODIS comparisons at these sites are shown in Tables 5 and 6, respectively.

About half of the matchups at COR, and about 2/3 at NSA, are mixed-phase clouds. Neither OLCI nor MODIS retrievals include a mixed-phase cloud type, meaning that overall skill at phase identification is poor. If the mixed-phase clouds are

**Figure 8.** Expected OLCI-ARM COT discrepancy vs. 68<sup>th</sup> percentile of CTH error (see text), for comparisons with (a) MFRSR and (b) Sun photometer date. Data are split into single-layer (blue) and multi-layer (red) cases. Numbers in parentheses indicate counts for each category. The dashed black line indicates 1:1.

**Table 5.** OLCI vs. ARM cloud phase confusion matrix. The rightmost two columns indicate the fraction of phase classifications agreeing at each site, first for all matchups, and then excluding points identified by ARM as mixed-phased conditions.

| Site | Satellite | AF     | ARM phase |       |      | Fractional agreement |  |  |
|------|-----------|--------|-----------|-------|------|----------------------|--|--|
|      | phase     | Liquid | Ice       | Mixed | All  | Non-mixed            |  |  |
| COR  | Liquid    | 11     | 2         | 9     | 0.40 | 0.07                 |  |  |
|      | Ice       | 0      | 2         | 3     | 0.48 | 0.87                 |  |  |
| NSA  | Liquid    | 128    | 39        | 363   | 0.26 | 0.70                 |  |  |
|      | Ice       | 12     | 63        | 127   | 0.26 | 0.79                 |  |  |

excluded, though, accuracies improve significantly. Both satellite data sets are significantly more likely to label mixed-phase clouds as liquid than ice, although note that they are most sensitive to the phase near the top of the clouds (which could be either phase dependent on circumstances). Misclassification of ice clouds as liquid is also more common than the converse.

## 4 Comparison against MODIS and SLSTR retrievals


To take a bigger-picture look at the CHROMA retrieval, we processed the OLCI record globally for the month of July 2019. The choice of month was somewhat arbitrary but at the time we began the analysis, only limited SLSTR retrievals were available. We subsampled OLCI at  $3\times3$  pixel resolution, taking the median reflectance from cloudy pixels (if any in the box) as input, to decrease computational overhead as the CHROMA-OLCI code is research-level and not optimized for speed. The implementation was otherwise as described in Section 2.1. We then applied the quality filtering described there, and aggregated the results on a 1° grid on a daily basis. We averaged COT geometrically and because it varies over several orders of magnitude.

**Table 6.** As Table 5, except for MODIS.





| Site | Satellite     | AF       | RM ph    | ase        | Fractional agreeme |           |
|------|---------------|----------|----------|------------|--------------------|-----------|
|      | phase         | Liquid   | Ice      | Mixed      | All                | Non-mixed |
| COR  | Liquid<br>Ice | 29<br>0  | 3 4      | 10<br>11   | 0.58               | 0.92      |
| NSA  | Liquid<br>Ice | 288<br>1 | 30<br>95 | 729<br>360 | 0.25               | 0.93      |

and CTP and CTH arithmetically. We focus on CTP arithmetically. The rationale for using CTP over CTH here is that this is the native coordinate of the CHROMA retrieval; unlike the ARM comparison above (where ARM natively provides CTH and can be considered a reference truth), none of the satellite data sets are a true reference for cloud altitude. We do provide some CTH statistics as a point of reference.

We aggregated SLSTR cloud retrievals from this month in the same way. We also use the MODIS standard daily cloud product (MOD08\_D3), which is already provided on this grid(and includes a geometric mean COT), and includes equivalent averaging methods to the one we chose for OLCI. In the following analysis, we only consider grid cells (on a daily basis) where all three instruments sampled with at least 100 satellite pixels on each day, and discard cells with a solar zenith angle above 70°. This gives a total of 761,875 common grid cells over the month.

Some summary statistics comparing the three satellite data sets (from this co-sampled, gridded data) are given in Table 7. We also show a mapped comparison between the data sets (focusing on differences relative to OLCI) in Fig. 9. This portion of the analysis is an intercomparison rather than a validation as the three different satellite retrievals have their own strengths and limitations and there is no reference known truth. For this reason, as with the ARM COT analysis, we use terminology of offset and difference as opposed to bias and error. Additionally, pixel-level uncertainties and mission goals do not translate directly to such gridded aggregates. We do, however, calculate the fractions of grid cells agreeing within PACE mission goals of 25 % for liquid COT, 35 % for ice COT, and 60 mb for CTP (F<sub>25</sub>, F<sub>35</sub>, F<sub>60</sub> respectively) as a general point of reference for magnitude discrepancies between the data sets. Prior to discussing differences, we first feel it is worth mentioning that examining daily and monthly aggregates for all three sensors (not shown) reveals they all show the same basic spatial and temporal features.

For COT, OLCI and SLSTR agree with one another better than either do with MODIS, on all metrics in Table 7 except median offset. The fact that these sensors are on the same satellite probably contributes to this - although MODIS on Terra has a similar overpass time it has a different orbital cycle and, even restricting to grid cells seen on the same day as here, the clouds might not be seen at the exact-same times as viewed by OLCI and SLSTR on Sentinel-3A. Thus there is additional discrepancy expected due to spatiotemporal variability in the clouds seen. Unfortunately, MOD08 does not provide averaged observation times so it is not possible to filter based on Terra vs. Sentinel-3A time difference using these files. A related factor is that high latitudes are relatively overrepresented in the sample due to orbital characteristics. Additionally, the MODIS cloud product is known to have some across-track patterns in retrieved properties. For example, in the previous MODIS Collection 5, Maddux

**Figure 9.** Cloud properties and differences for co-sampled grid cells by the three sensors in July 2019. Panels (a) and (d) show geometric mean OLCI COT and arithmetic mean OLCI CTP, respectively. Panels (b) and (c) show the median ratio of OLCI/MODIS and OLCI/SLSTR COT, respectively. Panels (b) and (c) show the median OLCI-MODIS and OLCI-SLSTR CTP, respectively. Grid cells with fewer than three days contributing are shaded grey.

et al. (2010) found (their Fig. 2) a tendency for higher liquid and ice COTs near nadir (view zenith angles up to 10° compared to near the edge of swath (view zenith angles of 60° or higher). Although that analysis concerned an older version of the MODIS algorithm, the underlying reasons - a combination of retrieval assumptions and the fundamental physical differences (both in sensitivity and what is observed) between viewing clouds from near nadir and at oblique angles - likely persist in some form, and similar phenomena likely affect the OLCI and SLSTR retrievals as well.




The primary factors influencing the satellites' reported COT include differing sensitivity of the underlying cloud masks (affecting the population of clouds on which retrievals are performed), radiometric calibration, and appropriate retrieval assumptions (chiefly surface albedo and cloud scattering properties, which are determined by CER and, for ice, shape and roughness). For liquid clouds midvisible wavelengths are less sensitive to assumed particle size than SWIR ones (Nakajima and King, 1990), so the fact that CHROMA assumes a CER is likely not a significant factor. For ice crystals, CHROMA and MODIS assume the same ice crystal habit model of severely roughened 8-element column aggregates from Yang et al. (2013). ORAC uses the generalised habit mixture of Baum et al. (2014), which has a more complicated CER-dependent asymmetry parameter than the column aggregates habit and may be one of several contributors to differences.

Over water (where surface albedo is low) OLCI COTs tend to be slightly lower than MODIS and SLSTR, although are a little higher than MODIS in a few areas such as the intertropical convergence zone (ITCZ) and some storm tracks. In many cases, differences are within  $\pm 25$  %. Lower values could be attributable to OLCI's cloud mask detecting fewer thin clouds (e.g.

**Table 7.** Summary statistics for comparison between OLCI, MODIS, and SLSTR co-sampled daily gridded data from July 2019. The rightmost two columns indicate the fraction of phase classifications agreeing at each site, first for all matchups, and then excluding points identified by ARM as mixed-phased conditions. Offsets are defined by the first sensor minus the second sensor. MAD and RMSD indicate median absolute difference and root mean square difference, respectively.

| Sensor pair F  |             | $F_{25}$    | $F_{35}$ | Median offset (%)         | MAD (%)  | RMSD (%)  |
|----------------|-------------|-------------|----------|---------------------------|----------|-----------|
| COT comparison |             |             |          |                           |          |           |
| OLCI/MODIS     | 0.73        | 0.40        | 0.51     | -11.7                     | 33.7     | 59.8      |
| OLCI/SLSTR     | 0.82        | 0.41        | 0.54     | -19.0                     | 31.6     | 54.4      |
| MODIS/SLSTR    | 0.78        | 0.40 0.52   |          | -7.3                      | 33.0     | 55.5      |
| Sensor pair    | $R_{\rm S}$ | F           | 60       | Median offset (mb) MAD (r |          | RMSD (mb) |
| CTP comparison |             |             |          |                           |          |           |
| OLCI/MODIS     | 0.72        | 0.          | 35       | -2.06                     | 90.2     | 161.3     |
| OLCI/SLSTR     | 0.82        | 0.          | 47       | -7.35                     | 65.3     | 116.0     |
| MODIS/SLSTR    | 0.84        | 0.52        |          | -7.21                     | 56.4     | 125.1     |
| Sensor pair    |             | $R_{\rm S}$ |          | Median offset (km)        | MAD (km) | RMSD (km) |
| CTH comparison |             |             |          |                           |          |           |
| OLCI/MODIS     |             | 0.58        |          | -0.42                     | 1.22     | 2.75      |
| OLCI/SLSTR     |             | 0.74        |          | -0.03                     | 0.96     | 1.84      |
| MODIS/SLSTR    |             | 0.74        |          | 0.27                      | 0.79     | 2.14      |

cirrus or fragments near cloud edges) than the other sensors (thus pushing up the average COT), due to OLCI's lack of SWIR or TIR bands which are useful for detection of optically-thin clouds. This seems particularly plausible in the ITCZ, where cirrus is common, and OLCI CTP is also higher (suggesting lower clouds, on average) than the other sensors. COT differences are more complicated over land; the patterns of COT offset with respect to MODIS and OLCI are moderately similar, suggesting different surface albedo assumptions between the assorted satellite algorithms contribute. One commonality is OLCI retrieving much lower COT over Greenland, which is likely due to difficulties in cloud masking and surface albedo modeling over this bright surface (which is also observed in fairly low-Sun conditions). MODIS COT is intermediate between OLCI and SLSTR.



For CTH and CTP, OLCI agrees better overall with SLSTR than MODIS for all summary statistics (except again median offset, in the case of CTP). The spatial patterns of CTP difference between OLCI and the others are quite similar with different magnitudes. This makes sense because MODIS and SLSTR are both TIR-based algorithms while OLCI uses the O<sub>2</sub> A-band so the two have differing fundamental physical and algorithmic sensitivities. MODIS' CO<sub>2</sub>-slicing channels should provide additional precision for the retrieval of high-level clouds beyond that of SLSTR (both sensors share TIR channels near 11 and 12 tmum). Additionally, the MODIS CTP retrieval is independent from the MODIS COT data set (different algorithms), while




**Figure 10.** (a) Mean fraction of clouds detected as multi-layer by MODIS. Grid cells with fewer than three days contributing are shaded grey. (b) Binned OLCI vs. other CTP difference as a function of MODIS multi-layer fraction. MODIS (blue) and SLSTR (red) data are offset horizontally slightly for clarity. Diamonds and lines show bin median and central 68 % of data, respectively; the zero line is dashed grey. In both panels cases, data are shown for co-sampled grid cells by the three sensors in July 2019.

OLCI and SLSTR use a single algorithm to retrieve both parameters simultaneously. CTP differences are larger in magnitude in the OLCI-MODIS comparison than the OLCI-SLSTR, especially over south-eastern Asia and the ITCZ. There are offsets of both signs, reinforcing the earlier discussion that A-band based cloud altitude retrievals are not necessarily always lower than those determined from TIR techniques.

We also examined how COT and CTP differed differ between the data sets as a function of A-band surface albedo (not shown). There were no large-scale patterns, aside from a large tendency for OLCI to retrieve lower COT than the others over the brightest surfaces (as visible in the Arctic and Greenland in Fig. 9). This is unsurprising due to the difficulties in cloud detection and accurate modeling of surface characteristics over the bright snow and ice.

As discussed previously, one key difficult situation with CTP retrievals, for which A-band and TIR techniques are expected to behave in different ways, is in multi-layer cloud systems. Figure 10 shows the average MODIS multi-layer cloud fraction (defined as the fraction of all clouds for which the MODIS data product indicates that a multi-layer cloud system is likely; Marchant et al., 2020) and the binned difference between OLCI and other CTPs as a function of the multi-layer fraction from the individual daily grid cells. Many of the areas with higher average multi-layer fraction are also ones of more positive OLCI CTP offset in Fig. 9, suggesting that differing sensitivities to multi-layer systems are important. This could also explain some of the COT offsets, due to differing phase identification leading to different cloud scattering properties being used (and in each case the single-layer not representing the multi-layer scattering properties well). Although the ITCZ is visible in this map, the multi-layer fraction is typically low (under 0.2) implying that other factors such as cloud detection are more significant here. The binned data in Fig. 10b show a roughly linear variation in OLCI CTP offset as a function of MODIS multi-layer fraction, more positive for MODIS than for SLSTR. This behaviour is consistent with our simulated multi-layer retrievals in Sayer et al. (2023), and the near-zero offset for multi-layer fractions under 0.1 confirms that the A-band and TIR techniques behave similarly in single-layer situations.

Another factor influencing the CTP retrievals more than COT is the cloud vertical extent assumptions. The TIR-based MODIS and SLSTR retrievals assume a cloud with negligible vertical extent; as the cloud is a grey body, in these situations if the cloud is opaque with high extinction (in the TIR) near the top then the retrieved cloud altitude will be very close to the cloud top. If the cloud is low COT then the retrieved altitude will be some point within the cloud (Sayer et al., 2011). Note that the visible and TIR COT are often similar for ice clouds, but TIR COT is generally lower than visible for water clouds (see Fig. 8 of Sayer et al., 2011). In contrast, as seen earlier, CHROMA assumes a finite geometrical thickness and its error is dependent on how appropriate that is. These differences between approaches may contribute to the negative CTP offsets in the low-COT parts of tropical oceans in Fig. 9, where the TIR retrievals may penetrate further into the cloud.

A complex issue influencing all of the above, in addition to differing accuracies of underlying cloud masks, is different failure rates of the COT and CTP retrievals between the data sets. This occurs when the algorithms are unable to find solutions meeting their internal consistency and quality checks. For CHROMA, manual examination of cases excluded by the iterations and cost thresholds described in Section 2.1 suggests that many of these are areas where the cloud mask was either a false positive, or the surface albedo assumption was incorrect. Cho et al. (2015) examined liquid-phase MODIS COT and CER retrievals over oceans and found higher failure rates for broken cumulus than stratocumulus overall; conditions associated with failure included inhomogeneity/sub-pixel cloudiness, Sun glint or large solar/view angles, and cloud mask or phase determination errors. MODIS CTP retrieval failures are less well-documented. Restricting the comparison to grid cells where MODIS and SLSTR cloud fractions are both within 0.05 of OLCI removes over 80 % of the sample; statistical metrics of the type shown in Table 7 do not appreciably change for COT or CTP when this is done (not shown), and as the resulting spatial sampling is different and quite limited it is not clear how meaningful this is in terms of understanding the level of agreement between the data sets at these gridded scales.

## 5 Conclusions and implications for PACE OCI






In this analysis we applied the CHROMA algorithm, developed primarily for cloud altitude retrievals for the OCI instrument on NASA's PACE mission, to the OLCI sensor on the Sentinel-3A satellite. We performed the bulk of the analysis prior to PACE's launch in February 2024; the purpose of the study is to evaluate and understand how the algorithm performs on real data before a large PACE validation data set can be acquired.

Comparing the OLCI-derived CTH to ground-based ARM reference data revealed a comparable quality to MODIS, with differences between the records generally consistent with physical understanding. OLCI retrievals have a tendency to underestimate the CTH as the true CTH increases, especially for multi-layer cloud systems. OLCI CTH uncertainty estimates have skill for single-layer opaque clouds although are too small because of the systematic nature of this bias. We anticipate continuing this type of validation exercise with PACE and ARM, and expanding to use similar data products from the Cloudnet network (Illingworth et al., 2007) as well.

We also compared the COT retrievals against ARM data from two instruments (for liquid phase clouds): MFRSR and Sun photometry. These two ARM data streams had around a 30 % offset from one another on average, which is problematic to

validate satellite retrievals, although the common data volume was fairly small and limited to two of the seven ARM sites used. OLCI and MODIS COT retrievals were closer to the MFRSR data, and uncertainty estimates (for single-layer clouds) seemed both skillful and reasonably well-calibrated. However, it is possible that both the MFRSR data and satellite retrievals respond to deviations from an idealised cloud in similar ways (e.g.—inhomogeneity leading to errors of the same sign) as they are more sensitive to this than Sun photometry often is.





Finally, we used ARM cloud phase data and found that both OLCI and MODIS performed significantly better than guesswork for discriminating between pure liquid or ice clouds. They were more likely to classify mixed-phase clouds as liquid than ice. These were, again, unfortunately only available for two of the sites used.

Our previous simulated retrievals suggested that, with a common algorithm configuration, OLCI and PACE OCI should perform similarly well (Sayer et al., 2023). As a result, we expect these results should be broadly applicable to OCI. One exception is that, with a 1 pm local overpass time compared to Sentinel-3's 10 am, OCI will see lower solar zenith angles. Potentially, this means smaller 3D effects (less self-shadowing by clouds), which is an advantage. However, photons would also pass through a slightly smaller amount of atmosphere, and see slightly larger scattering angles, and it is unclear whether this is a small advantage or disadvantage on average. The ARM analysis suggests several avenues for development which could, particularly for OCI, improve the retrieval:

- 1. Investigate more advanced (beyond Voigt) O<sub>2</sub> absorption line shapes, and update line data bases and continuum absorption models as they become available, as the spectral contrast in absorption drives the retrieved cloud altitude.
- 2. Reassess the assumptions about cloud geometric depth (or attempt to add this as a retrieval parameter, even though it would likely not be robust in some circumstances). The evaluation supported our previous simulated retrievals suggesting that this is an important factor contributing to retrieval bias. This is expected to be a more significant driver of retrieval quality than O<sub>2</sub> line models, based both on previous sensitivity studies, and the fact that O<sub>2</sub>-based CTH retrievals have been performed for decades (see e.g. the introductory discussion in Sayer et al., 2023).
- 3. Consider adding a mixed-phase cloud type, or developing a flag for such cases, as they can be common in some regions and would likely be associated with poor COT retrievals (although this could not be tested with the ARM products we used).

OCI has some capabilities beyond OLCI which could assist with the above, such as measurements in the O<sub>2</sub> B-band (although these may saturate in bright scenes) and H<sub>2</sub>O absorption channels (which could provide additional constraints if the H<sub>2</sub>O profile is known). Additionally, OCI's SWIR channels should improve on cloud masking and phase determination beyond the VIS-NIR spectral range of OLCI. Nevertheless, multi-layer clouds are likely to remain problematic for passive retrievals; in. In principle combining O<sub>2</sub> absorption with TIR bands, such as the OLCI-SLSTR combination, could prove fruitful; the combination of these spectral regions has been explored recently by Nagao et al. (2025). PACE is not co-mounted with a TIR radiometer but has orbital overlap with MODIS, VIIRS, and geostationary sensors which could be used for a subset of their mutual records.

We found the three-way OLCI, MODIS, and SLSTR COT/CTP-comparison to provide broadly consistent conclusions to the ARM analysis, although one key difficulty (related more broadly to the use of satellite spatiotemporal aggregates) is that, due to differing orbital characteristics, sensor/algorithm sensitivities, and retrieval failure rates, sampling is incomplete and can be biased. We did not use the Cloud-Aerosol Lidar and Infrared Pathfinder Satellite Observations (CALIPSO) CALIPSO lidar and CloudSat radar missions in this analysis, even though they can act as a spaceborne reference for cloud altitude and phase (Stephens et al., 2018), because these flew in the A-train in an early afternoon orbit and observations were not spatiotemporally coincident with the morning Sentinel-3A and Terra overpasses. The CALIPSO and CloudSat missions have now ended, but in May 2024 the Earth Cloud, Aerosol and Radiation Explorer (EarthCARE) mission, with similar active sensing capabilities, launched in a similar early-afternoon orbit. PACE is also in an early-afternoon Sun-synchronous polar orbit, and their tracks intersect in the Southern hemisphere, meaning EarthCARE will be able to be used as a source of frequent validation for CTH (albeit over a limited latitude range) once both data streams are routine.






COT evaluation is a more challenging problem; the two ARM data streams were well-correlated with each other, and with MODIS and OLCI retrievals, although systematically offset. These offsets should ideally be better understood but could plausibly linked to factors such as spatial scales of cloud heterogeneity. PACE's polarimeters (Martins et al., 2018; van Amerongen et al., 2019) will provide independent COT retrievals though these have their own uncertainties. Ice COT is a particular challenge due to the effects of variations in crystal size, shape, and roughness, which can vary greatly both within and between clouds, on scattering and absorption. The NASA PACE Postlaunch Airborne eXperiment (PACE-PAX, https://espo.nasa.gov/pace-pax) took place in September 2024 around coastal California (USA) and collected data which will be highly valuable to validate PACE cloud properties. However, as field campaigns are unavoidably limited in time and space, additional work will be necessary to evaluate and improve our understanding of space-based cloud remote sensing. A common thread through all of these comparisons is that the fundamentally different nature of ground-based and satellite observations - in terms of observation geometry, field of view, and spectral range used to probe the clouds - makes evaluation of satellite cloud retrievals a challenging task.

Code availability. The CHROMA retrieval code as eventually applied to OCI will be available within NASA SeaDAS (https://seadas.gsfc.
 nasa.gov/) once implemented as a standard product within the PACE data processing system. The research codes used for the implementation here and validation analysis are in IDL and available from the lead author upon request.

Data availability. The OLCI retrievals are available from the author upon request as they are not a NASA standard product. OLCI RSRs are available at https://dragon3.esa.int/web/sentinel/technical-guides/sentinel-3-olci/olci-instrument/spectral-response-function-data. The TROPOMI surface data base is available at https://www.temis.nl/surface/albedo/tropomi\_ler.php. MERRA2 data are available from https://gmao.gsfc. nasa.gov/reanalysis/MERRA-2/. ARM data are available from https://adc.arm.gov/discovery, as data streams arsclkazr1kolliasC1, mfrsr-cldod1minC1, sphotcod2chiuC1, and thermocldphaseC1. MODIS cloud products are available from https://ladsweb.modaps.eosdis.nasa.gov/, and SLSTR from https://cds.climate.copernicus.eu/datasets/satellite-cloud-properties/ with citation Copernicus (2022).

Author contributions. AMS led development of the CHROMA algorithm, performed OLCI data processing, performed the analyses, and led preparation of the manuscript. GET provided the SLSTR satellite data used. SG and DZ are leads of several of the ARM VAPs used. All authors provided advice during algorithm development and/or evaluation, and contributed to editing and review of the manuscript.

*Competing interests.* Authors AMS and LL are Associate Editors for Atmospheric Measurement Techniques. This manuscript was handled by an independent editor.

Acknowledgements. NASA-affiliated authors were funded by the NASA PACE project. LL was funded by the Alexander von Humboldt foundation via the Feodor-Lynen fellowship 2020. This work (SEG and DZ) was also supported by the US Department of Energy (DOE) Atmospheric Radiation Measurement (ARM) program and Atmospheric System Research (ASR) program. This paper has been authored by an employee of Brookhaven Science Associates, LLC, under contract no. DE-SC0012704 with the US DOE. René Preusker (Freie Universität Berlin) and Julien Chimot (EUMETSAT) are thanked for the OLCI smile distortion model, as well as numerous discussions on OLCI data. Christine Chiu's (Colorado State University) insights into the Sun photometer and MFRSR COT data were greatly appreciated. NASA's Global Modeling and Assimilation Office (GMAO) are thanked for the MERRA2 meteorological data used as ancillary input for the CHROMA algorithm. We acknowledge the free use of the TROPOMI surface DLER database provided through the Sentinel-5p+ Innovation project of the European Space Agency (ESA). The TROPOMI surface DLER database was created by the Royal Netherlands Meteorological Institute (KNMI); Gijsbert Tilstra (KNMI) is thanked for assistance in better understanding this data base. Ground data were obtained from the ARM user facility, a U.S. DOE Office of Science user facility managed by the Biological and Environmental Research Program. Site staff, algorithm developers/VAP translators, and the ARM program are thanked for the creation and stewardship of these data records - in particular, in addition to those on the author list of this manuscript, Karen Johnson and Dié Wang (Brookhaven National Laboratory).

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
