# Peer review of "Evaluation of cloud height, optical thickness, and phase retrievals from the CHROMA algorithm applied to Sentinel-3 OLCI data"

_EGUsphere, 2025_

## Author Comment (AC1)

We are grateful to the four reviewers for their positive comments on our study, and suggestions for revision. This document contains our responses to all four reviewers. Below, reviewer comments are in *italics* and our responses are in regular type. In addition to this, in response to a request from the Editor, we have extended the *y*-axis range on Figure 3.

**Referee 1**

*The authors have made a thorough, comprehensive yet clearly explained evaluation of OLCI CHROMA cloud parameter retrievals.*

*I recommend publication with only minor remarks.*

Thank you for your comments and recommendation.

*Introduction. The authors have used ARM ground-based data but not ACTRIS Cloudnet ground-based data (https://cloudnet.fmi.fi/, Illingworth et al. (2007)), although this network delivers very similar data, among which CTH and cloud phase data, and is also used for satellite cloud product validation (e.g., Compernolle et al., 2021, Vinjamuri et al. (2023)). Please provide some rationale why ACTRIS Cloudnet was not considered. Please also include a paragraph with references on how ARM and Cloudnet was previously used in satellite cloud product validation.*

We agree with the value of Cloudnet and will be including it in the routine validation of OCI cloud top height retrievals, together with ARM. This study with OLCI data was based on a research version of the CHROMA code – not integrated into NASA's operational data processing systems (because the OLCI application is a research project in support of PACE OCI work as opposed to a dedicated NASA product). There are therefore more computational limitations on how much OLCI data we are able to store and process on our research machine (there is considerable overhead). We considered both networks but chose to use ARM for this study because (1) it has a manageable number of sites which are more globally distributed than Cloudnet, which is largely clustered in Europe (so the sites in ARM can be considered more independent samples) and (2) several ARM sites provide MFRSR and Sun photometer liquid COT data, which we use, and to our knowledge Cloudnet does not provide these quantities. We judged that the use of ARM data alone would be sufficient to verify the overall performance of CHROMA and assess whether this is broadly in line with our previously-published (Sayer *et al.* 2023) study based on simulated retrievals. We feel that we do indeed have enough ARM-OLCI matchups to make these judgements and support the conclusions of our study.

In the revised manuscript, we have expanded Section 2.2 and the Conclusions to discuss these choices and mention previous CTH validation efforts from ARM and Cloudnet. This includes several of the references the reviewer suggests citing, and others.

*Page 6, line 148. Please provide some numbers how the CTP 60 mb uncertainty goal translates to CTH uncertainty. This is currently provided on page 13, line 273, but it should (also) go here.*

This is a good suggestion, and we have done so in the revised manuscript.

We would also like to note that we felt this paragraph fit better in the "matchup methodology" section than in the "ARM data section" because it was more about how we look at the matchups as opposed to the ARM data itself.

*Page 6, line 148. Please provide reference for standard atmospheric profile.*

We've added a reference here in the revised manuscript.

*Page 6, line 149. Where do these numbers for COT uncertainty goals 25%, 35% come from? Also from Werdell et al 2019? Please indicate.*

Yes, this is correct – the Werdell paper is already cited in this sentence for this purpose, although we've moved the location of the citation in the sentence so it is more directly associated with these numbers.

*Page 7, line 181 and page 8, line 1. 'the retrieved CTH'. Retrieved by the satellite, right? Please make more clear by stating 'where h is the CTH retrieved by the CTH'*

Yes, this is correct – we have made it explicit in both cases in the revised manuscript.

*As indicated by the authors, the CTH retrieved by the satellite can have an error, in which case the parallax correction will also have an error. Have the authors considered to combine the CTH of ARM (which is the ground-truth) with the viewing zenith angle of the satellite to estimate the correction? Is there a reason why this was not considered?*

We did think about doing it this way, but there are several reasons that we chose not to. There is no perfect solution unless you have a nadir overpass of the site simultaneous with the ground measurement. The concerns basically arise from the inhomogeneity of the cloud field(s) over the domain – variations in height, presence of multiple cloud layers, etc. We see from the matchups and from looking at individual satellite overpasses that this can be common and non-negligible. We want to estimate the true lat/lon of clouds for every pixel in the satellite image in the vicinity and due to this heterogeneity it's easiest to use the satellite CTH as an input for that (as this is a first-order accounting for variation around the site) as opposed to the single-location instantaneous ground CTH. The downside there is the uncertainty in satellite CTH gives uncertainty in the true lat/lon. The alternative would be to take the ground-based CTH and satellite geometry and back out the pixel which sees the cloud directly above the ground site at the time. However, there's the danger of multi-layer clouds where there is another layer which is seen by the satellite – that is, the pixel seeing the cloud above the site might also be seeing other clouds which are not above the site. So it's not straightforward. In short, if the cloud field is homogeneous then it should not matter, and if it's variable then neither approach is ideal.

We have not noted this in the revised manuscript as we feel that additional discussion would break the flow. However as the reviewer comment and our responses to them remain publicly viewable and citable after publication, we feel that documenting this in the Response to Reviewers is enough.

*Table 1 and 2. This table has 3 main categories: 'COT<3', 'single-layer COT>=3', 'multi-layer COT>=3'. However, from fig 3 it is clear that there is also a strong dependence on CTH. I would therefore propose to have within each category, below each row with 'All', a row for low clouds (e.g., CTH<3 km or CTH<4 km) and a row for high clouds (CTH >3 km or > 4 km).*

We prefer not to expand the tables because there are already quite a number in the manuscript and these ones are quite large (and we worry not so readable as is). We framed the analysis in the three categories for programmatic reasons: those are the ones we are required to use to evaluate PACE OCI data with so we need the point of reference. We have, however, calculated some of these metrics for subsets of ARM CTH below and above 3 km and expanded the discussion in section 3.2 and 3.3 to include them where relevant, so the reader gets some of this information.

*Page 12, line 255-256. "while the ARM data may be sensitive to lower cloud droplet concentrations (and retrieve a higher top) than satellite remote sensing, this would not account for a multi-km error"*

*In Sneep et al. (2008), figure 1, large differences are obtained between the cloud pressure from MODIS based on thermal infrared, and those based on retrieval using the O2 A absorption, using O2-O2 absorption and rotational Raman scattering.*

*Given these large differences, is it therefore not imaginable that the CTH negative bias of OLCI CHROMA is because it is based on O2 A absorption? Note also that in the S5P TROPOMI Cloud routine validation, large discrepancies are obtained between the cloud top height from OCRA\ROCINN CAL (based on O2A absorption) and that of Cloudnet (Lambert et al., 2025, table 2).*

We don't believe that $O_2$ A-band absorption inherently means that there will be a negative bias in retrieved CTH. These are physically based retrievals involving radiative transfer which is understood well. We realise that many (A-band and otherwise) do have a bias in common. Rather a likely common cause is that some assumptions in the algorithms themselves contribute to this bias. Two main common assumptions, whose effects are typically conflated, are: (1) assumed cloud model (2) assumed cloud vertical structure.

(1) Assumed cloud model

The referee points to Figure 1 in Sneep *et al* (2008). The $O_2$ A-band products, displaying CTH underestimation, plotted in that Figure are described in the paper as follows:

   *[13] Both OMI cloud products use basically the same cloud model, which is the same as that*

*used in FRESCO [Koelemeijer et al., 2001]. The cloud is represented by a Lambertian surface with albedo 0.8; that is, no light is transmitted through the cloud.*

The consequence of the assumption of a Lambertian cloud is that it is not possible for the algorithm to account for multiple scattering inside the clouds. If photon penetration within a cloud is neglected, any increase in oxygen absorption due to multiple scattering is interpreted by the algorithm as a cloud at a level that is lower that the actual altitude. This can be seen in Figure R1. The CTHs retrieved by the FRESCO algorithm, using the same cloud Lambertian model of the algorithms of Figure 1 in Sneep *et al* (2008), are low biased with respect to the true cloud boundaries (blue lines). Those retrieved by the SACURA algorithm, equipped with a scattering cloud model, are closer to the true cloud boundaries. This is because multiple scattering inside the cloud is accounted for by computing the gaseous single scattering albedo within the $O_2$ A absorption channel (see Eq. 8, page 1554, in Lelli *et al.*, AMT, 2012).

[Figure]

**Figure 2.** FRESCO-retrieved cloud heights for simulated cloud cases for a solar zenith angle of 60° and nadir view. The cloud optical thickness values are $\tau_1 = 5$, $\tau_2 = 15$, $\tau_3 = 35$, $\tau_4 = 60$. Cloud top and base heights are indicated using blue horizontal lines. The surface albedos ($A_s$) are 0.3 and 0.05.

Wang, P. and Stammes, P.: Evaluation of SCIAMACHY Oxygen A band cloud heights using Cloudnet measurements, Atmos. Meas. Tech., 2014.

[Figure]

Lelli, L., Kokhanovsky, A. A., Rozanov, V. V., Vountas, M., Sayer, A. M., and Burrows, J. P.: Seven years of global retrieval of cloud properties using space-borne data of GOME, Atmos. Meas. Tech., 5, 1551–1570, https://doi.org/10.5194/amt-5-1551-2012, 2012.

**Figure R1:** Literature examples of cloud top heights from (left) Lambertian and (right) full scattering cloud descriptions in $O_2$-based retrievals.

(2) Assumed cloud vertical structure

Our retrieval simulation study (Sayer *et al.*, 2023) indicated that an error in assumed fractional geometric depth (FGD) could cause a bias of this type (because of photon penetration through the clouds), and our analysis of ARM FGD from the matchups in the current paper shows that the true FGD distributions differ from our assumptions in the way that would lead to biases consistent with those we see in the retrievals. We see this effect in Figure R2, where CTH retrieved by the $O_2$ A-based CTH SACURA, ROCINN, and CHROMA algorithms are collocated with CALIOP CTH for TROPOMI orbit 04691 (this figure is unpublished work led by co-author L. Lelli). All $O_2$ A-band based retrievals assume a fixed liquid water content profile. This introduces a bias common to all three algorithms, which is particularly noticeable for clouds at altitudes between 4 and 10 km. SACURA determines CTH and CBH simultaneously, showing a much

less pronounced bias. Note that the TROPOMI application of CHROMA is an initial test with a simplified spectral response function so not fully reflective of OCI/OLCI performance.

[Figure]

**Figure R2:** Comparisons between CTHs from four algorithms (three TROPOMI O2 A-band based and VIIRS thermal) for one scene. Colour indicates distance between the CALIOP track and ground point. The 1:1 (grey) and regression (pink) lines, and some comparative statistics, are shown.

Many other (A-band and otherwise) also make simplifying assumptions about cloud vertical structure – for example, that clouds are geometrically thin layers. For example OMI retrieves an effective pressure (often called "optical centroid") for a homogeneous cloud while we attempt to retrieve the top under the assumption that vertical extent is known. So we feel that there is a physically based cause which can in principle be addressed here. Of course this is not the only source of errors so addressing this may not fully remove this bias, but we do not believe it is hopeless or inherent to the technique.

We also note that various other methods (using either $O_2$ bands or $O_2$ and thermal in concert) do show skill at retrieving cloud base and top heights and that these do not seem to show the same negative bias tendency in CTH (e.g. SACURA in Figure R2). It is a factor of information content of measurements – of knowing when the measurements alone are sufficient to determine what is desired, vs. when they are not, knowing what the appropriate external assumptions and constraints to impose on the retrieval are. Here we clearly did not make the optimal assumptions, and now that we have been able to assess this using real observations it will assist in refining the algorithm. The comparisons with MODIS and SLSTR in Figure 9 show regional offsets of both signs, further strengthening the view that it is not a case that $O_2$ CTH retrievals are necessarily lower than those from TIR or active techniques.

Section 3.2 discussed the bias and likely contributing factors (e.g. FGD assumptions, possible limitations of spectroscopy used). In the revised manuscript we have expanded the discussion here with some examples from the literature.

*Page 20, line 379. The authors motivate why they use CTP instead of CTH. Still it would be good to have results for CTH as well, as it gives a point of intercomparison with the OLCI vs*

*ARM comparisons. In particular, it would be interesting to have a figure similar to Fig 3 (CTH difference vs CTH) for OLCI CHROMA vs MODIS and vs SLSTR, to check if discrepancy increases with CTH.*

We gave these points long consideration. Neither MODIS nor SLSTR gridded retrievals can be considered the truth so presenting such a plot needs careful framing, plus the sampling from global gridded data is very different to the instantaneous site-based matchups. So we worry that, whatever the results, such a plot might be overinterpreted with respect to the equivalent OLCI-ARM plot in Figure 3. Plus, this would further extend the manuscript (which was not short and has had a fair amount of additional discussion added in this revision).

As a compromise, we have generated Tukey mean-difference (sometimes also called Bland-Altman) plots for CTP and CTH from this co-sampled level 3 data, and include those below (Figure R3). This keeps them viewable and citable by an interested reader and provides some of the extra context from the review which might be lost or damage the flow if it were added into the manuscript. These show that there is some CTH and CTP dependence of the difference, but it varies somewhat between SLSTR and MODIS, the patterns are not linear, and the secular changes are smaller than the variability within bins. Additionally, in the revised manuscript we explain the CTH vs CTP rationale more and provide CTH statistics in Table 3 (note that unlike COT or CTP there is no equivalent simple $F_{60}$ metric that could be added).

[Figure]

**Figure R3:** Tukey mean/difference plots comparing OLCI and SLSTR or MODIS CTP (left) and CTH (right). As in the main body of the paper, plots are generated using daily level 3 data from July 2019 where all three sensors provided sufficient retrievals. Data binned in 20 equally-populated bins. Symbols indicate bin median values and lines indicate the central one standard deviation of differences.

**Referee 2**

*This paper introduces the CHROMA algorithm which was developed to retrieve cloud information such as cloud height and optical thickness. As the authors explain, this algorithm was intended to be used for the OCI instrument on the new PACE mission. The algorithm was applied to OLCI data instead. The results presented in this paper are therefore also interesting to the OLCI community.*
*The topic of this paper is certainly fitting for AMT. The paper is interesting to read and well written. I think this paper deserves publication.*
*I came across a few minor typos:*
*- page 6, line 125: "although is" -- this should probably be something like "although it/this is"?*
*- page 7, line 172: "0.1°resolution" --> "0.1° resolution" (space missing)*

Thank you; we are glad you find the paper useful. We have corrected both of these typographical issues in the revised manuscript.

**Referee 3**

*This paper presents a comprehensive and timely evaluation of the CHROMA algorithm's cloud products (CTH, COT, and phase) derived from Sentinel-3 OLCI data. The study effectively uses ground-based ARM data and intercomparisons with other satellite products to assess the algorithm's performance, providing guidance for the future application of CHROMA to PACE OCI data. The methodology is sound, the analysis is thorough, and the paper is well-written and clearly structured. This work will be of great interest to the community and is well-suited for publication in AMT. I recommend its publication with only a few typos I identified.*

*Minor Corrections*

- *Line 39: 'select' -> 'selected'*

- *Line 116: 'KAZRASRCL' is a typo and should be 'KAZRARSCL'*

- *Line 217: The phrase "implying larger outliers" is slightly awkward. Suggest rephrasing to "which implies the presence of larger outliers..."*

- *Line 289: 'counfound' is a typo for 'confound'.*

- *Figure 9 Caption: 'arithmeric mean' should be corrected to 'arithmetic mean'.*

Thank you; we are glad you find the paper useful. We have corrected these five typographical issues in the revised manuscript.

**Referee 4**

*This paper is well-written and covers an important topic. The authors validate cloud property retrievals from OLCI, including a validation of their uncertainty. I was not very familiar with oxygen-based cloud top height retrievals myself, and found the paper interesting to read. I have a few clarification questions, but otherwise it can be published with minor revisions.*

Thank you for your comments and recommendation – it is reassuring that our work is interesting and understandable to someone less familiar with $O_2$-based CTH retrieval. We have addressed these comments as indicated below.

*§2.1, page 4, line 97: by "spectrally correlated uncertainty", do the authors mean an error covariance?*

Yes, we represented as an error covariance matrix (with perfect correlation) in that work. For clarify, we've added this term in parentheses in the revised version.

*§3.1, page 8, line 195: the phrasing in this sentence is a bit confusing. You mean you use medians when analysing the difference? And you don't only use medians, your tables also contain RMSE. I think this sentence would fit better in §3.2.*

We use medians instead of means when doing the spatiotemporal matching (which is why the text is here). We have added expanded the text here in the revised version to make this clearer. In the statistical analysis later we also prefer medians instead of means (for some of the same reasons) but yes we use other metrics as well.

*§3.2, page 8, line 211: I suggest to replace "disproportionately" with "strongly", as I don't know in proportion (or disproportion) to what NSA would be overrepresented, or "disproportionately overrepresented" is a pleonasm here.*

We have deleted the word "disproportionately" as we agree it wasn't necessary and think "overrepresented" is sufficient. The main point here was that the number of matchups is unevenly distributed between sites and NSA had the most by a large margin, because it is overflown more frequently than the other sites.

*§3.2, table 1, page 9: The score for multi-layer GUC for F_60 and F_ED is very low at 0.00.*

Yes, this is true. The algorithm, as expected and discussed, performs poorly in multi-layer cases and it seems particularly poorly at this site (although the data volume is low). There does not appear to be a request for an edit here so no change has been made in the manuscript. As the review and response remain online and viewable/citable we think that acknowledging this in the response is sufficient.

*§3.2, page 12, line 262: "it is not clear how differences from Voigt would be integrated", could this be estimated?*

We cannot easily estimate because the required beyond-Voigt line shape parameters were not available in HITRAN at the time the analysis took place and are not available at the time of writing (HITRAN2024 release has been delayed). We had otherwise hoped that HITRAN2024 would have been released and there would have been time to assess and/or update. The alternative would be to take in a different gas absorption data base which includes these and develop a different code base to regenerate some simulations. Given this would be a lot of work and we do not expect that this is the biggest factor in the retrieval error we felt it was most appropriate to acknowledge that this was a potential error source. Now that PACE is on-orbit, once HITRAN2024 is available we plan to update and test the effects of beyond-Voigt line shapes with OCI data.

*§3.2, page 13, line 268: fixed although FGD is not robustly retrievable, couldn't you do better than a fixed value? For example, a fixed value per cloud type?*

Yes, this is discussed later (Figure 5 and lines 280-290 of the submitted manuscript). This is something we are planning to test with OCI data, now we have evidence from the real retrievals in this study, supporting our previous work with simulated data, that it is a factor.

*§3.2, page 14, figure 5a: I find the results in this figure surprising. At COT between 10 and ~60, an increase in CTH is associated with an increase in FGD. But FGD is defined as the vertical cloud fraction between cloud top and surface. Shouldn't that mean that FGD should be expected to be close to 1 for very low clouds?*

It depends on how close the cloud base is to the ground. It is also important to bear in mind that the results are dependent on the mix of sites available for matchups. An increasing FGD with CTH for moderate/high COTs like this implies convective development for many of the cases sampled. If a cloud starts with a top at 1 km and base at 0.5 km, this gives a FGD of 0.5. If the same cloud develops so the top is now 4 km and base is still 0.5 km, FGD is now (4-0.5)/4 = 0.875.

*§3.2, page 14, line 289: "not be globally representative", the authors could consider if A-Train or EarthCare data can be used. You comment on this in the conclusions, but it could be mentioned in the beginning why this was not possible for this study.*

We prefer to leave this in the conclusion as we feel that extending the introduction would detract from the flow of the manuscript. We feel it fits better in the conclusion because that section is also more forward-looking to PACE OCI, where EarthCARE is available (and we have begun to explore this following the recent release of EarthCARE retrieval products). We also note that EarthCARE will not provide globally representative matchups because its orbit moves in the

opposite direction from other early-afternoon polar orbiting satellites (daytime descending vs. daytime ascending), which is why instantaneous matchups are restricted in latitude range.

*§3.3, page 15, figure 6: I understand why uncertainty estimates are poor for multi-layer clouds, and this is commented on in the text. But why are they so poor for COT<3, apparently even worse than for multi-layer clouds?*

We think that this relates to a topic discussed in Section 3.2: that sometimes low optically thin liquid or mixed phase clouds are retrieved as ice clouds (especially at NSA). This puts them around 3 km (the minimum ice CTH allowed by the retrieval LUT used). So they are positively biased but the retrieval thinks it has found a good fit, hence the uncertainty estimate is too small. Our previous study with simulated data noted non-robustness of phase retrieval for low COT. In the revised manuscript we have expanded the discussion here to better signpost this.

*§3.3, page 15, lin 313: uncertainty estimates are not skillful. Could this be improved?*

Yes; in the revised manuscript we have expanded the discussion around here in response to a few reviewer comments.

*§3.4, page 18: those F_ED values are very low, in particular for MFRSR matchups. This is true even for single-layer clouds. Are the uncertainy estimates overconfident? But this does not show from Figure 8b? The table and figure seem to show conflicting messages about the reliability of the uncertainty estimates.*

$F_{ED}$ for OLCI COT for single-layer clouds in Table 3 is 0.65 and 0.67 overall (for sun photometry and MFRSR respectively), very close to the expected value (~0.68), indicating reliable results, and consistent with the OLCI results shown in Figure 8. We do not see an inconsistency in the message here. We wonder if the reviewer is mistakenly looking at Table 4, which shows MODIS results, where the corresponding fractions are 0.19 for sun photometry and 0.16 for MFRSR?

*§3.5, page 19, line 365: As the authors note, the phase comparisons may not be generalisable. This makes them hard to use. I'm not sure if they should be included in the paper.*

We prefer to keep them because, even if not generalisable globally, they represent the extent to which this quantity can be validated by ARM at the present time. We feel it is important to document this and the results can be a point of reference for the future. Additionally, we feel the numbers may be useful to some potential data users because Arctic clouds are a research focus for some scientists studying cloud radiative effects and feedbacks. The paper does note that the results are likely not globally representative, and the section is short (two paragraphs and one table), so we are not devoting too much of the paper to them.

*§3.5, page 20, line 370: I am surprised that satellite data sets are more likely to label mixed-phase clouds as liquid than as ice, even though they are sensitive near the top where ice should be relatively dominant.  Why does this happen?*

We can't say for sure but it might be related to their altitude (the current CHROMA retrieval only allows for liquid phase clouds below ~3 km). Many of these clouds are also optically thin, for which the cloud contribution to the signal (especially over a bright surface) is small.

*§4, page 20, line 379: Why is COT averaged geometrically but CTP arithmetically?  I would understand arithmetic for CTH, but CTP drops exponentially with CTH, so I would think a geometric mean to be more appropriate.*

We averaged COT geometrically because the natural distribution is closer to lognormal than normal. Geometric COT, arithmetic CTP, and arithmetic CTH are also all provided as standard in the MODIS level 3 products we used. So this is also a matter of the common practice in the community.

*§5, page 24, line 480: OLCI CTH uncertainty estimates only have skill for COT > 3 single layer clouds.  For others they don't have skill.  This line promises a bit too much.*

We agree and have added the words "for single-layer opaque clouds" here in the revised manuscript.

*§5, page 25, line 509: Note that OLCI also has $H_2O$ absorption channels at 0.91 µm.  Could they be used if the $H_2O$ profile is known?*

Yes, in principle – this was something we briefly discussed in our prior study (Sayer et al 2023). We chose not to pursue it in this initial development of CHROMA partially because OLCI and OCI's spectral characteristics around this region are not as close as they are for A band measurements. OLCI has discrete bands at 885, 900, 940, and 1020 nm with 10, 10, 20, and 40 nm FWHM, respectively. OCI's hyperspectral portion (5 nm FWHM) ends around 895 nm, it lacks a band near 900 nm, its 940 nm band has a 44 nm FWHM, and it has a band around 1040 nm with 74 nm FWHM. So while the A-band measurements make OLCI a good proxy, incorporating the $H_2O$ measurements risked overestimating how well the method would translate to OCI, and also would entail a much more significant code development effort. That's the reason we opted to focus only on the $O_2$ A-band first, and once performance for that is established, assess the expected value of adding B-band measurements (which OLCI lacks so could not be tested) and $H_2O$ from various channels. The revised manuscript signposts these future options a bit more.

*Editorial:*

*line 289: counfound → confound*

*line 345: remove "to"*

*line 431: missing ) after μm and μ should not be italic*

*line 435: differed → differ*

Thank you; we have fixed all of these in the revised manuscript.